# Multi-Omic Meta-Analysis of Transcriptomes and the Bibliome Uncovers Novel Hypoxia-Inducible Genes

**DOI:** 10.3390/biomedicines9050582

**Published:** 2021-05-20

**Authors:** Yoko Ono, Hidemasa Bono

**Affiliations:** Program of Biomedical Science, Graduate School of Integrated Sciences for Life, Hiroshima University, 3-10-23 Kagamiyama, Higashi-Hiroshima, Hiroshima 739-0046, Japan; d205302@hiroshima-u.ac.jp

**Keywords:** hypoxia, RNA-seq, ChIP-seq, gene2pubmed, bibliome, meta-analysis, signature genes, *GPR146*, enrichment analysis

## Abstract

Hypoxia is a condition in which cells, tissues, or organisms are deprived of sufficient oxygen supply. Aerobic organisms have a hypoxic response system, represented by hypoxia-inducible factor 1-α (HIF1A), to adapt to this condition. Due to publication bias, there has been little focus on genes other than well-known signature hypoxia-inducible genes. Therefore, in this study, we performed a meta-analysis to identify novel hypoxia-inducible genes. We searched publicly available transcriptome databases to obtain hypoxia-related experimental data, retrieved the metadata, and manually curated it. We selected the genes that are differentially expressed by hypoxic stimulation, and evaluated their relevance in hypoxia by performing enrichment analyses. Next, we performed a bibliometric analysis using gene2pubmed data to examine genes that have not been well studied in relation to hypoxia. Gene2pubmed data provides information about the relationship between genes and publications. We calculated and evaluated the number of reports and similarity coefficients of each gene to HIF1A, which is a representative gene in hypoxia studies. In this data-driven study, we report that several genes that were not known to be associated with hypoxia, including the G protein-coupled receptor 146 gene, are upregulated by hypoxic stimulation.

## 1. Introduction

The development of genetic engineering techniques such as genome editing have made it possible to test hypotheses that are constructed on the basis of the knowledge of each researcher, including information in the literature, by performing biological experiments. Hypothesis construction is triggered by literature surveys and the collective knowledge of researchers, and is also influenced by publication bias. For example, when the frequency with which approximately 600,000 publications annotated around 20,000 human coding genes was calculated, more than 9000 publications reported the p53 gene whereas over 600 genes were not mentioned at all [1]. In contrast, the development of microarray and high-throughput sequencing technologies has enabled the comprehensive acquisition of thousands of gene expression profiles at a time. Published transcriptome data are archived in public databases such as the Gene Expression Omnibus (GEO) of the U.S. National Center for Biotechnology Information (NCBI, Bethesda, MD, USA) [2], ArrayExpress (AE) of the European Bioinformatics Institute [3], and the Genomic Expression Archive (GEA) of the DNA Data Bank of Japan [4]. At present, although approximately 60,000 expression data series of human genes have been registered, we estimate that they are rarely used in studies. We believe that meta-analysis of these transcriptome data may provide additional insights into biology.

Among the various fields of biological research, the more well-studied the field, the greater the impact of publication bias. We considered areas where meta-analysis based on gene expression data can uncover new findings buried by publication bias. The analyses were performed by considering the following conditions: (1) hypoxia-inducible factor 1-α (HIF1A) is a representative transcription factor involved in hypoxic stimulus-response, and (2) categorization in Gene Ontology, a database where gene expression data can easily be obtained. In our previous study, we studied hypoxic stimulus-responses using public databases [5]. Here, we performed a meta-analysis of gene expression data following the previous method. The present study is a continuation of our previous study to further build our knowledge of the hypoxic stimulus-response. In a previous report, we integrated meta-analyzed hypoxic transcriptome data with public ChIP-seq data of known human HIFs, HIF-1 and HIF-2, to gain insight into hypoxic response pathways involving direct binding of transcription factors. Due to publication bias, there has been little focus on genes other than the well-known, signature hypoxia-inducible genes. Therefore, in this study, we conducted an omic-scale analysis of bibliological data in PubMed. Bibliometric analysis is a method of analyzing biological publications and it is used in biomedical research [6]. This new type of analysis is called “bibliome, and it involves using NCBI’s gene2pubmed data, which provides information about the relationship between genes and publications, to discover novel hypoxia-inducible genes.

Oxygen is essential for respiration in aerobic organisms. Hypoxia occurs in a variety of neurological conditions such as traumatic brain injury, Alzheimer’s disease, and stroke, and is known to be responsible for some of their symptoms. In hyperbaric oxygen therapy, subjects are placed in a chamber filled with 100% oxygen gas at a pressure of 1 atm or higher, and it is mainly used to treat hypoxia-related symptoms [7]. Oxygen is used to produce ATP by oxidative phosphorylation and electron transfer systems. Hypoxia is a condition in which cells, tissues, or organisms are deprived of sufficient oxygen supply as the amount of oxygen in the blood decreases. The tissues and cells of higher organisms, including humans, have a hypoxic response system that is regulated by HIFs to adapt to the oxygen-deprived conditions caused by hypoxic stimuli.

Under normoxic conditions, HIFs are hydroxylated by the α-ketoglutarate-dependent dioxygenase factor inhibiting HIF-1 (FIH-1) and prolyl hydroxylases (PHDs), resulting in degradation by the ubiquitin–proteasome system and suppression of transcriptional activation [8,9]. However, under hypoxic conditions, the activity of PHDs and FIH-1 is reduced, and HIFs escape hydroxylation and form a complex with aryl hydrocarbon receptor nuclear translocator (ARNT) and the transcriptional co-factor CREB-binding protein to induce expression of downstream genes [10].

In this study, we focused on hypoxic stimulus-response and aimed to identify novel genes by performing meta-analysis using public databases. We selected hypoxia-inducible genes based on the gene expression ratio of approximately 500 pairs of hypoxia-stimulated samples and evaluated the similarity coefficient of each gene for HIF1A, a representative factor in hypoxia research, using gene2pubmed, which shows the relationship between genes and publications.

## 2. Materials and Methods

### 2.1. Curation of Public Gene Expression Data

To obtain the accession of hypoxia-related gene expression data from public databases, we used a graphical web tool called All of Gene Expression (AOE) [11,12]. AOE integrates metadata from not only GEO, AE, and GEA, but also RNA sequencing (RNA-seq) data that are exclusively archived in the Sequence Read Archive (SRA) [13]. We searched for a list of experimental data series related to hypoxia from GEO using the following search formulas: “hypoxia” [MeSH Terms] OR “hypoxia” [All Fields] AND “Homo sapiens” [porgn] AND “gse” [Filter]. We downloaded this list on 17 August 2020. We obtained the metadata from the Series Matrix files in GEO, from articles, or downloaded the metadata from SRA using the Python package pysradb (v 0.11.1). We then curated only the RNA-seq data into comparable sample pairs of hypoxia and normoxia (HN-pairs) [14]. The criteria for curation were that the specimens with the experimental conditions of hypoxia and normoxia should be in the same data series so as to set the HN-pair and to derive RNA-seq reads from human cell lines or tissue specimens. All relevant data were adopted if the metadata content was certain. We collected the SRA format data from the NCBI using prefetch (version 2.9.6).

### 2.2. Gene Expression Quantification

Since the downloaded data were in the SRA format, we used the fasterq-dump program in SRA Toolkit [15] to convert the data into FASTQ formatted files for expression quantification. We then quantified single-end and paired-end RNA-seq reads using ikra (version 1.2.3) [16], an RNA-seq pipeline with default parameters. Ikra automates the RNA-seq data analysis process, which includes the quality control of reads (Trim Galore version 0.6.3) and transcript quantification (Salmon version 0.14.0 [17] with reference transcript sets in GENCODE release 30). We chose this workflow because it is suitable for quantifying gene expression from heterogeneous RNA-seq data obtained from various instruments and laboratories. In this study, the data acquisition and quality control processes required approximately 1 month to obtain the current dataset. Quantitative RNA-seq data are accessible at figshare [18].

### 2.3. Calculation of HN-Ratio and HN-Score

Based on the transcriptome data paired with hypoxia and normoxia, we calculated the ratio of expression value for each gene (termed the HN-ratio) [19]. The HN-ratio *R* is calculated using the following equation:(1)$$R=\frac{T_{hypoxia}+1}{T_{normoxia}+1}$$
where $T_{hypoxia}$ is the gene expression value described as scaled transcripts per million (scaledTPM) [20] under hypoxic conditions and $T_{normoxia}$ is scaledTPM under normoxic conditions, paired with hypoxic conditions. To reduce the effect of small variations in extremely low gene expression values, we calculated the HN-ratio by adding 1 to the denominator and numerator.

We then classified each gene into three groups based on its HN-ratio. When the HN-ratio was greater than the threshold, the gene was considered upregulated, and when the ratio was less than the inverse of the threshold, the gene was considered downregulated. If a gene was neither up- nor downregulated, it was classified as unchanged. For the classification of up- and downregulated genes, we tested 1.5-and 2-fold thresholds, and then selected the 1.5-fold threshold to classify upregulated and downregulated genes. For evaluating hypoxia-inducible genes, we calculated the HN-score for each coding gene in humans. The HN-score was calculated by subtracting the number of samples with downregulated genes from the number of samples with upregulated genes.

### 2.4. Enrichment Analysis

We used Metascape [21,22] and ChIP-Atlas [23,24] for Gene Set Enrichment Analysis. ChIP-Atlas is a comprehensive and integrated database for visualization and utilization of publicly available chromatin immunoprecipitation sequencing (ChIP-seq) data. In this study, we performed conventional “express analysis” using Metascape. In the ChIP-Atlas settings, we set the “Select dataset to be compared” item to “Refseq coding genes (excluding user data)” and set the other items to default.

### 2.5. Meta-Analysis of ChIP-Seq Data

Public ChIP-seq data were collected, curated, and pre-calculated for reuse in the ChIP-Atlas database [23,24]. In ChIP-Atlas, the category of proteins that bind to DNA is referred to as “antigens”. We used the “Target Genes” function of ChIP-Atlas to obtain average model-based analysis of ChIP-seq (MACS2) scores for three antigens at a distance of ±5k from the transcription start site. These antigens were selected from the following genes that were shown to be related to the UP 100 genes by the ChIP-Atlas enrichment analysis: HIF1A, endothelial PAS domain-containing protein 1 (EPAS1, also known as hypoxia-inducible factor-2α), and ARNT. We combined the ChIP-seq data with the HN-score data described above, using the gene name to combine the two datasets.

### 2.6. Calculation of the Number of Publications for Each Gene and Similarity Coefficient for HIF1A

We calculated the number of publications and Simpson similarity coefficient per human gene using Python (version 3.8.6) [25]. The Simpson similarity coefficient *S* was calculated using the following equation:(2)$$S\left(X,Y\right)=\frac{|X\cap^{\;}Y|}{min\left(|X|,|Y|\right)}$$
where $|X\cap^{\;}Y|$ is the number of PubMed IDs that overlaps between HIF1A and a gene in gene2pubmed, whereas min(|X|,|Y|)  is the number of PubMed IDs of HIF1A or the gene in gene2pubmed, whichever value is less. Gene2pubmed [26] and Gene Info [26], which were required for the above calculations, were downloaded on 4 January 2021. In order to select a gene to determine whether or not hypoxia-related studies have been reported, we calculated the similarity coefficients of each gene for EPAS1 and ARNT as well as HIF1A. After reviewing the results, we determined that HIF1A was suitable.

### 2.7. Visualization and Integrated Functional Analysis of Genes

To generate scatter and box plots, we used TIBCO Spotfire Desktop version 11.0.0 (TIBCO Spotfire, Inc., Palo Alto, CA, USA).

## 3. Results

### 3.1. Overview

This study was performed in three main steps (Figure 1). In Step 1, we obtained the RNA-seq data related to hypoxic stimulation from public gene expression databases. After the manual curation of metadata, UP 100 and DOWN 100 gene lists related to hypoxia stimulation were obtained through the quantification of gene expression in the corresponding datasets. The UP 100 and DOWN 100 gene lists included the top 100 and bottom 100 genes according to HN-score. In Step 2, we verified hypoxia-related genes in the UP 100 and DOWN 100 gene lists, not only by enrichment analyses, but also by visualizing the HIF-related ChIP-seq peak of each gene using HN-scores. Finally, in Step 3, we investigated novel hypoxia-inducible genes using bibliometric analysis of gene2pubmed.

### 3.2. Curation of Hypoxic Transcriptome Data in Public Databases

Initially, we used AOE to check whether a large number of hypoxia-related data were registered. We then curated the metadata provided by the NCBI for further analysis. Next, we obtained the SRA IDs for 69 data series and 495 HN-pairs of GEO-drawn samples after integration with hypoxia-related data lists provided in previous reports and elimination of duplicates [14]. This resulted in a four-fold increase in HN-pairs compared to previous reports. In 495 HN-pair data, the hypoxic conditions ranged between 0.1% and 5% O_2_ concentration, including some chemical hypoxic conditions where CoCl_2_ was used to induce a hypoxia-related state under normoxic conditions. The treatment time ranged from 1 hour to a maximum of 3 months. The most common hypoxic condition among the data was 1% O_2_ (266 HN-pairs, 53.7%) and 24 h of treatment (234 HN-pairs, 47.3%). The most common cell type was cancer (324 HN-pairs, 65.5%), and the most common tissue was breast cancer (112 HN-pairs, 22.6%).

### 3.3. Meta-Analysis of Hypoxia-Inducible Genes

After the quantification of gene expression data obtained from RNA-seq, the number of conditions under which each gene was upregulated, downregulated, and remained unchanged was estimated on the basis of the HN-ratio. The complete lists of meta-analyzed results are accessible from figshare [27]. We calculated the HN-score of each gene to evaluate the responsiveness of each gene toward hypoxia. For example, in the case of vascular endothelial growth factor A (*VEGFA*), the counts showed 406 UP, 25 DOWN, and 64 unchanged, and therefore, its HN-score was 381. Full lists of counts (upregulated/downregulated/unchanged) with HN-scores for all genes are accessible from figshare [27]. We selected the top 100 [28] and bottom 100 [29] HN-scored genes and used these genes as the UP 100 and DOWN 100 gene lists, respectively, for further analysis.

### 3.4. Evaluation of Hypoxia-Inducible Genes

To confirm whether the UP 100 and DOWN 100 gene lists were affected by hypoxia-related regulators such as HIF1A, we performed enrichment analysis using ChIP-Atlas [23,24]. Enrichment analysis in ChIP-Atlas showed that a set of genes in the UP 100 gene list was related to hypoxia-related antigens such as HIF1A, ARNT, and EPAS1, whereas a set in the DOWN 100 gene list was related to epigenetic regulators such as Sin3A associated protein 30 (SAP30), histone deacetylase 1 (HDAC1), and MYC proto-oncogene, bHLH transcription factor (MYC), an oncogene involved in cell cycle progression (Table 1). Similarly, enrichment analysis by Metascape showed that hypoxia-related gene sets and genes related to the metabolic processes of non-coding RNAs (ncRNAs) were listed in the UP 100 and DOWN 100 gene lists, respectively (Figure 2a,b). Since it was important that the hypoxia-inducible genes were correctly evaluated by HN-score as intended in this analysis, we confirmed this not only by ChIP-Atlas enrichment analysis, but also by visualizing the high HIF-related MACS2 score of each gene with a high HN-score. The scatter plots of the mean MACS2 peaks of HIF1A, EPAS1, and ARNT, which were the top-ranked genes in the enrichment analysis of the UP 100 gene list, were plotted. The genes with higher MACS2 peaks also had higher HN-scores (Figure 2c,d). These results indicate that the genes regulated by hypoxia-related factors are listed in the UP 100 gene list, and genes regulated by SAP30, MYC, and HDAC1 are listed in the DOWN 100 gene list.

### 3.5. Evaluation of Simpson Similarity between HIF1A and Genes

We performed bibliometric analysis, a method of analysis using biological publication information, to determine whether the genes of the UP 100 gene list had been previously studied in relation to hypoxia. We calculated the number of publications for each gene in the UP 100 gene list [30]. We also calculated the Simpson similarity coefficient between these genes and HIF1A [31]. The number of publications and Simpson similarity coefficients were visualized using a scatter plot (Figure 3a). In addition, we defined genes that were reported to be involved with HIF-1 15 years ago [32] as “Classical HIF-1 regulated genes”, and these were marked on the scatter plot. Interestingly, those classical hypoxia-inducible genes were plotted on the upper side of the regression line.

### 3.6. Box Plot of HN-Ratio by Treatment Time

To check the individual HN-ratio of genes with high HN-scores based on the treatment time, we visualized base-2 logarithm (log2) transformed HN-ratios using the box plot (Figure 3b). Genes in the box plot were selected from the HIF-1-regulated gene *VEGFA,* and some genes were plotted below the regression line, such as sperm-associated antigen 4 (*SPAG4*), G protein-coupled receptor 146 (*GPR146*), protein phosphatase 1 regulatory subunit 3G (*PPP1R3G*), transmembrane protein 74B (*TMEM74B*), and serine protease 53 (*PRSS53*). The six genes visualized in the box plot showed positive HN-ratio values for the majority of the treatment time.

## 4. Discussion

In this study, we identified hypoxia-inducible genes by performing metadata analysis. We reported that several genes that were not known to be associated with hypoxia, *GPR146, TMEM74B*, *PPP1R3G*, and *PRSS53*, were upregulated by hypoxic stimulation.

We analyzed 69 data series related to hypoxia, which were selected from the entire gene expression data registered in GEO, and obtained 495 HN-pairs, which is about four times more than previously reported [5]. In this study, 65.5% of the gene expression data were derived from cancer experiments, reflecting the demand for these data in hypoxia-related cancer research and ease of handling. We believe that each gene was evaluated without publication bias. Our study was possible because of the availability of data from public databases.

First, the Metascape and ChIP-Atlas enrichment analyses of the UP 100 gene list showed their association with hypoxia-related gene sets as expected, and hence, we concluded that HN-score helps select the hypoxia-inducible genes (Table 1, Figure 2). Thus, we decided to proceed with the analysis of the UP 100 gene list in subsequent analyses. On the other hand, Metascape enrichment analysis of the DOWN 100 gene list suggested an association with the metabolic processes of ncRNA (Figure 2b). This suggests that hypoxic stimulation may also affect the expression of ncRNAs. In this study, only the protein-coding genes of GENCODE were quantitatively analyzed, and hence, ncRNAs were excluded from this analysis. Further studies are needed to identify ncRNAs involved in hypoxic stimulation.

Gene2pubmed has been used to determine the order of genes retrieved by RefEx [33]. In this study, we used gene2pubmed to evaluate the genes that are not as well-known as hypoxia-related genes. We used gene2pubmed to calculate two variables: the number of publications per gene and the Simpson similarity coefficient for HIF1A. The Simpson similarity coefficient measures the strength of the co-occurrence of PubMed IDs associated with HIF1A or with a gene. If the Simpson similarity coefficient is close to 1, the relationship between PubMed IDs and HIF1A is considered strong; however, the lower the number of PubMed IDs on the side being compared indicates a poor relationship. For this reason, we visualized the number of publications and Simpson similarity coefficient for the genes in the UP 100 gene list in the scatter plot. In addition, we also plotted HIF-1-regulated genes [32] in this plot (Figure 3a). The Simpson similarity coefficient was lower in the case of more publications; this might be because of the fact that more relationships were revealed, and thus, the Simpson similarity coefficient is likely to be underestimated. The classical HIF-1 regulated genes scored high in the Simpson similarity coefficient and was plotted on the upper side of regression line, despite a large number of publications. The genes with a small number of publications, such as ankyrin repeat domain 37, which was reported to be regulated by HIF1A [34], were also plotted above the regression line. Therefore, we hypothesized that the genes plotted below the regression line, such as *GPR146*, *SPAG4, TMEM74B, PPP1R3G* and *PRSS53* might be considered as candidates for novel hypoxia-inducible genes. The log2-transformed HN ratio of these genes was visualized as a box plot (Figure 3b). These genes had positive HN-ratio values, as well as genes with known hypoxic responses, such as *VEGFA.* Although some reports state that *SP**AG4* is related to hypoxia [35,36], no reports were available in a PubMed search related to hypoxia-related genes other than *SP**AG4*.

We focused on GPR146 because it is a G protein-coupled receptor and therefore worth considering as a drug target and because it has a higher HN-score than other genes. In the similarity coefficient analysis using gene2pubmed, three reports on the co-occurrence of PubMed IDs between HIF1A and GPR146 were available [37,38,39]. These were overall genetic analyses and did not focus on hypoxia. We confirmed that *G**PR146* was upregulated, which was plotted below the regression line in the scatter plot (Figure 3a), similar to other known hypoxia-inducible genes. *GPR146* was not reported to be associated with hypoxia in the PubMed search; thus, we believe that *GPR146* is an important hypoxia-inducible gene that has not been focused on so far. In hypoxia studies using comprehensive gene expression data [40,41], *GPR146* is included in the list of genes upregulated by hypoxic stimulation. However, according to previous reports, *GPR146* is just a gene among many hypoxia-related genes. The inhibition of GPR146 is involved in lowering cholesterol levels [42]. C-peptide, a putative ligand of GPR146, inhibited low O_2_-induced ATP release in erythrocytes [43]. The biological function of GPR146 is expected to be further elucidated in future studies.

We focused on the hypoxic response and visualized the number of research reports for each gene using gene2pubmed data. We identified the novel genes by using public data containing thousands of pieces of gene expression information, which can be obtained regardless of the researcher’s interest. As shown in Figure 3b, the gene expression variation at each time point was not constant. Also, the tissues and cells in each data set were different. In this study, we only made a rough evaluation of the variation in gene expression by using the collective intelligence from the public database. Further stratification analysis will be undertaken in our future work. We plan to continue using public databases to discover similar new findings. For example, we believe that the ncRNA field, where hypothesis generation is difficult due to the lack of information, has great potential for new discoveries. At present, the number of known human long ncRNA transcripts exceeds several hundred thousand [44,45]. We plan to investigate the gene expression of ncRNAs under hypoxic conditions, based on our results of the effect of hypoxia on metabolic process-related genes of ncRNA (Figure 2b).

## 5. Conclusions

Multi-omics analysis of the transcriptome and bibliome revealed that several genes, which were not known to be associated with hypoxia, were upregulated under hypoxic conditions.

## Figures and Tables

**Figure 1 biomedicines-09-00582-f001:**
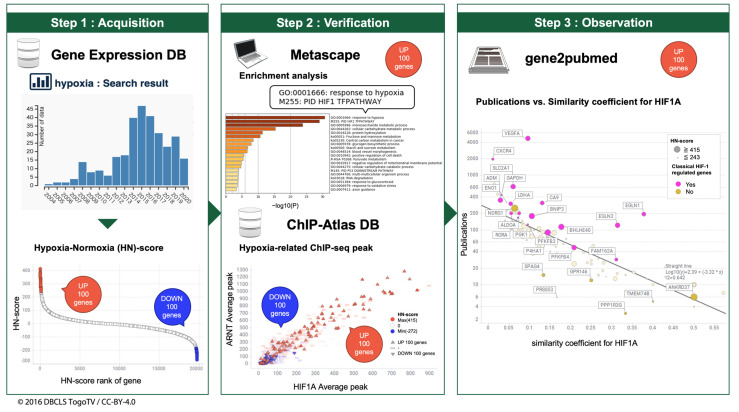
Schematic view of hypoxic transcriptome meta-analysis. Step 1. Evaluation and listing of upregulation and downregulation of hypoxia-inducible genes. Step 2. Confirmation of known hypoxic stimulation-related genes. Step 3. Discovery of novel genes related to hypoxic stimulus-response.

**Figure 2 biomedicines-09-00582-f002:**
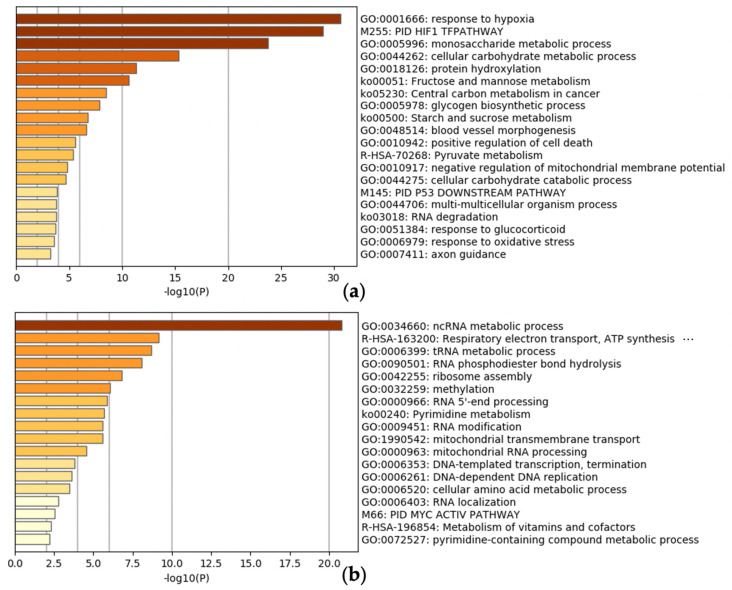
(**a**,**b**) Confirmation of known hypoxic stimulation-related genes. Enrichment analysis for (**a**) the UP 100 gene list and (**b**) the DOWN 100 gene list. (**c**,**d**) Scatter plot of ChIP-seq average peaks of hypoxic-related antigens, (**c**) HIF1A vs. ARNT and (**d**) EPAS1 vs. ARNT colored by HN-score.

**Figure 3 biomedicines-09-00582-f003:**
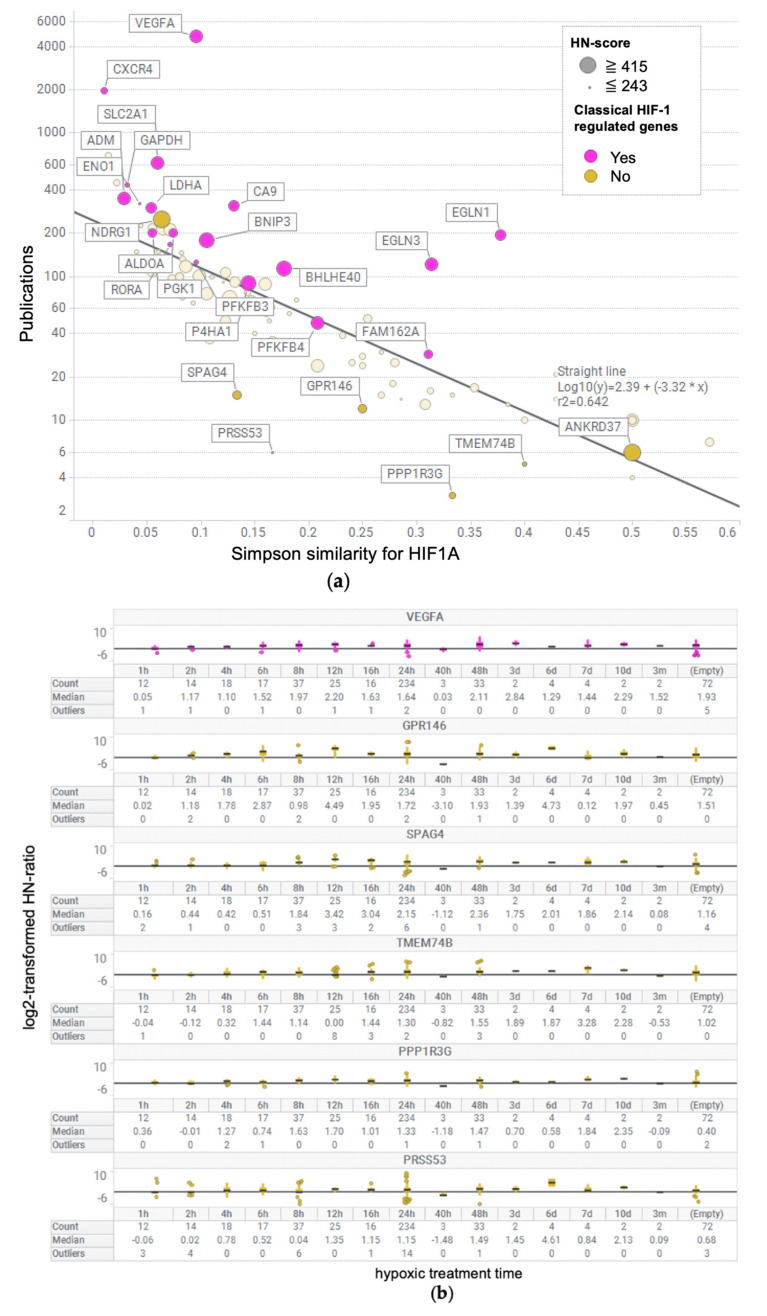
Discovery of novel genes associated with hypoxic stimulus-response. (**a**) Scatter plot of the number of publications vs. Simpson similarity coefficients for HIF1A in the UP 100 gene list. In this scatter plot, genes that were reported to be associated with HIF-1 15 years ago were marked as “known genes regulated by HIF-1”. (**b**) Box plot of log2-transformed HN-ratio per hypoxic treatment time for some of the UP 100 genes.

**Table 1 biomedicines-09-00582-t001:** The result of enrichment analysis in ChIP-Atlas. Enrichment analysis in ChIP-Atlas is a search tool for target genes and colocalizing factors of a given transcription regulator.

Input List	Antigen	ID	Log P-val	Log Q-val	Fold Enrichment
UP 100 gene list	HIF1A	SRX4802348	−88.8246	−84.064	35.6249
	ARNT	SRX4802353	−76.4303	−72.3686	83.3136
	EPAS1	SRX3051209	−73.1987	−69.2831	34.9928
DOWN 100 gene list	SAP30	SRX116447	−34.1844	−30.0149	4.96916
	MYC	SRX1497384	−31.4158	−27.5474	2.97453
	HDAC1	SRX186644	−27.4205	−24.1541	3.3231

## Data Availability

The data presented in this study are openly available in figshare [46]. Source codes to replicate the study are also freely available at GitHub under the MIT license [25].

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
