# Peer review of "Multi-Omic Meta-Analysis of Transcriptomes and the Bibliome Uncovers Novel Hypoxia-Inducible Genes"

_biomedicines, 2021, doi:10.3390/biomedicines9050582_

Round 1

Reviewer 1 Report

The manuscript by Ono and Bono used several publicly available transcriptome datasets for the identification of novel hypoxia-induced genes. In addition, they attempted to draw conclusions about happy hypoxia associated with Covid-19. The approach taken is good, but the actual analysis left out many things that are critical to the study of hypoxia and its pathways. The manuscript needs to be extensively revised. Below are a few general comments first.

General points

Meta-analysis: There are guidelines about how to perform a meta-analysis. You should follow these guidelines. For example, have you excluded some data set? Yes? Why. Have you used all data set for all the different analysis?

Covid: The Covid story appears from time to time in the manuscript, but is not mentioned in the title or abstract. Also, it is not really clear why this analysis was done. What the message is and why it was important. Added value? It rather distracts from the actual statement of the manuscript. Either it needs to be made clearer, or I would leave it out.

Hypoxia: The scientific approach is very limited, because many aspects are neglected, which would be much more interesting.

  1. Role of HIF2a is completely left out. It would be much more interesting, for example, to develop gene prolife, which makes it possible to distinguish between HIF1 and HIF2 target genes. Which ones are specific.
  2. The study also completely leaves out the fact that there are differences between long-term and short-term hypoxia. So I will not be able to compare gen profiles after 24h hypoxia with those after 5 days. There are studies that show that as an acute response HIF1a is upregulated and then HIF2a, while HIF1a falls again. This should not be considered in the study, or at least discussed.
  3. Moreover, a normal tissue cell reacts differently to hypoxia than a tumor cell. This should be taken into account, or at least discussed. Your data are very important and interesting, but they need to be analyzed more precisely and carefully.

Title: Why Multi-omics? The study only analyzed gene expression profiles. The title is misleading and needs to be changed. In addition, why focused on GPR146? The manuscript contains no additional data about GPR146. Its just one gene of the data set. Please rework the title carefully.

Abstract

Line 10: Avoid is not the correct word in this context

Line 10: Sentence needs to be rewritten “Because of publication bias,…”

Line 20: Again, why GPR146? Its not the only gene in your manuscript. Why is this one especially important?

What’s about EPAS1/HIF2α?

Introduction

Line 29-31: The two sentences about the hypothesis needs to be summarized.

Line 48-50: What’s the difference to your previous study? Have you used the same data set? This needs to be clarified.

Line 56-57: What´s about EPAS1/HIF2α? Again, avoid is not the correct word in this regard (adapt?) What do HIFs cause? Should be explained at least briefly.

The story with Covid and “happy hypoxia “comes out of nowhere at all points in the manuscript. It also has nothing to do with the actual story of the manuscript. It needs to be better introduced. Why are you doing it? What message are you hoping to make. What is the relevance of the investigation.

Methods

Line 124-126: “These antigens…” At various places in the manuscript, you use the word “antigens”. This is not correct in this context. Please have a look at the corresponding definition. Also, it is not clear in this sentence why you did what exactly. This passage needs to be carefully revised.

Is the analysis related to COVID somewhere explained? Please check and add if you decide to leave this analysis in the manuscript.

Results

Figure 1: Nice figure, but the legend needs to be improved. Covid is mentioned in the figure but not in the legend. Also

Section 3.5: The whole section needs clarification.

Figure 4: How have you defined “well known genes to be regulated by HIF1”? NDRG1 is for example well known

Discussion

Start with what is new. Which genes have you newly identified?

Line 254-261: “showed the association with MYC,…” How exactly do you come up with that? Is it HIF1 or HIF2? Needs to be better explained and differentiated. Also the connection with ncRNAs is not quite clear. Also insert reference there if necessary.

Line 265-272: Again, the connection and they statement to Covid is not entirely clear. Needs to be revised.

Conclusion

If the identification of GPR146 is your main finding. You should at least clearly the biological relevance. Why is your study useful? What is new? This needs to be explained more carefully.

Reviewer 2 Report

This manuscript is focused on the meta analysis of transcriptomes and finds GPR146 as highly involved in hypoxia. By doing so, this manuscript improves our understanding of hypoxia on the molecular level, and expand our knowledge beyond the well studied protein Hif1a. The use of publicly-available data sources is smart and enables further data mining from already published data sets. 

I have no major issues with this manuscript, and can only suggest the following suggestions:

  • The authors should elaborate more on the molecular players involved in hypoxic condition, on the cellular and molecular levels.
  • The authors should also refer to hypoxic states in neurological conditions such as neurodevelopmental disorders, and how HBOT, for example, can improve the hypoxic conditions.
  • Figure 1 is great. But please elaborate in the text referring to step2 of figure 1, mentioning also the ChIP-Atlas part. 
  • The fonts in the figures (mainly 2-4) are too small. The authors should increase the fonts to improve readability by all readers from all ages. 
  • It can be beneficial for the research community if in their discussion, the authors will raise ideas for further studies based on their current findings, as well as ideas for other published data-based research approaches.

Round 2

Reviewer 1 Report

Even though the first iteration has improved the quality of the manuscript considerably, there are still some critical points that the authors should address. I think the decision to remove the COVID-19 related data from the manuscript is a good one.

Introduction

Line 48-51: As already indicated, it is not really clear what is new about the study, especially in relation to previous studies by the authors. At this point, the authors should also explain what the results of the previous studies are, in order to work out what is new in the present one and to clearly delimit it.

Line 69-75: Add reference

Methods/Results

Are really just RNA-seq data included in your study? No microarray data? Make sure that this is consistent in the whole manuscript

Table 1: How is the downregulation reflected in the values presented? “Fold enrichment” perhaps a little bit misleading. To show the fold change is maybe the better way. Please explain also in the legend.

Why have you used such an old report as reference? In the last 15 years more and more HIF target genes were described. As explained during the last review, some of the genes are well know HIF1 related genes, but they are listed as not well known. You should update the list.

Discussion

Line:267-271: EPAS1 and HIF1 have a different influence on the MYC/MAX complex and regulate the expression of MYC target genes. Please read the literature again carefully and correct it according to the current state of knowledge.

Round 3

Reviewer 1 Report

Thank you for the revision of the manuscript. The revisions have significantly improved the manuscript.